# CROSS-MODAL RETRIEVAL AUGMENTATION FOR MULTI-MODAL CLASSIFICATION

## ABSTRACT

Recent advances in using retrieval components over external knowledge sources have shown impressive results for a variety of downstream tasks in natural language processing. Here, we explore the use of unstructured external knowledge sources of images and their corresponding captions for improving visual question answering (VQA). First, we train a novel alignment model for embedding images and captions in the same space, which achieves state-of-the-art image-caption retrieval performance w.r.t. similar methods. Second, we show that retrieval-augmented multi-modal transformers using the trained alignment model significantly improve results on VQA over strong baselines. ~~reporting state-of-the-art performance~~. We further conduct extensive experiments to establish the promise of this approach, and examine novel applications for inference time such as hot-swapping indices.

## 1 INTRODUCTION

Neural networks augmented with non-parametric retrieval components have recently shown impressive results in NLP (Khandelwal et al., 2019; Guu et al., 2020; Lewis et al., 2020; Izacard & Grave, 2020). In this work, we train a state-of-the-art image-caption alignment model and utilize it in various retrieval-augmented multi-modal transformer architectures, achieving ~~state-of-the-art performance on visual question answering (VQA)~~ significant improvement over the baselines, including the winner of the VQA 2.0 2020 challenge.

Retrieval components are promising because they allow for easy revision and expansion of their memory, as compared to their parametric, pre-training counterparts. They provide more interpretability, as well as direct factual consistency with trusted knowledge sources. In the multi-modal setting, retrieval augmentation allows for leveraging the strengths of text-based models—as evidenced by the strong performance of BERT-based models in vision-and-language (Lu et al., 2019; Li et al., 2019b; Kiela et al., 2019)—via cross-modal translation from images to text. Being able to seamlessly "hot swap" knowledge sources without the need for re-training the model affords a unique scalability not typically seen in the traditional deep learning literature. Nearest neighbor methods are known to be strong baselines in the vision and language domain (Devlin et al., 2015).

We introduce a simple yet effective novel dense cross-modal alignment architecture called DXR (Dense X-modal Retriever). DXR achieves state-of-the-art performance on both COCO (Chen et al., 2015) and Flickr30k (Young et al., 2014) image-caption retrieval, with respect to similar methods. We subsequently use DXR to augment several multi-modal transformer architectures with a retrieval component. We show that retrieval augmentation yields impressive results for a variety of well-known multi-modal transformer architectures, ranging from VisualBERT (Li et al., 2019b) and ViLBERT (Lu et al., 2019)—which use bounding-box features—to Movie+MCAN (Jiang et al., 2020)—which uses grid features. We name our overall method XTRA, for X-modal Transformer Retrieval Augmentation. Specifically, our contributions are as follows:

- We introduce a novel image-caption retrieval architecture, DXR, that achieves state-of-the-art performance on COCO and Flickr30k, with respect to similar methods.

- We introduce a new retrieval-augmented multi-modal transformer architecture, XTRA, that achieves ~~state-of-the-art performance~~ significant improvement on VQA over the baselines.

To our knowledge, this is the first work to showcase the promise of hybrid parametric and non-parametric models for the vision and language domain.

- We conduct extensive experiments to shed light on this novel approach. We explore different datasets for training the alignment model, as well as the effect of in-domain versus out-of-domain retrieval indices, the index size and inference time applications. Our experiments show that our proposed method significantly improves over a variety of strong multi-modal baselines, and demonstrates superior results over pre-training.

## 2 RELATED WORK

**Cross-Modal Retrieval**    Prior work in cross-modal retrieval can be divided into two primary categories: (i) methods that use grid-features and/or vector representations of the embedding space, and (ii) methods that use detection features, sequence representations, or share information between the two modalities for computing the similarity metric. The first category consists of methods such as RRF (Liu et al., 2017) and DPC (Zheng et al., 2017) which use two convolutional network branches for image and text. CMPM by Zhang & Lu (2018) introduced a pre-trained image backbone with a Bi-directional LSTM to learn image and text embeddings. The most relevant work in this category is VSE++ (Faghri et al., 2017), which focuses on hard negative mining and ranking loss. The second category generally exploits the use of detection features, which enforces an additional complexity. Methods such as TERN (Messina et al., 2020b), TERAN (Messina et al., 2020a), SAEM (Wu et al., 2019) and MMCA (Wei et al., 2020), use transformer modules to obtain modality-specific embeddings. TERAN, as well as SCAN (Lee et al., 2018), utilize sequence similarities. SCO (Huang et al., 2018) and VSRN (Li et al., 2019a) learn, in addition to image-text alignment, to generate the caption from the image embedding. MMCA, as well as CAMP (Wang et al., 2019), fuses image and text information to obtain the final embeddings. Other methods, such as Unicoder-VL (Li et al., 2020a), Oscar (Li et al., 2020b) and UNITER (Chen et al., 2020) learn to align between image and text by using positive and negative tuples of images and captions. While these models perform well, they suffer from high computational complexity as we discuss in Sec. 3.4

**External Knowledge Source Methods**    The use of an external knowledge source (KS) has gained much attention in the field of natural language processing (NLP), such as the work of Verga et al. (2020). Our work is inspired by that of Lewis et al. (2020), which introduced RAG, a generic approach for a variety of downstream NLP tasks, that uses a learned retriever (DPR by Karpukhin et al. (2020)) to augment the inputs by marginalizing across several retrieved phrases retrieved from Wikipedia. In the multi-modal domain, previous efforts have focused on building different types of KS, such as the work of Zhu et al. (2014), Chen et al. (2013), Divvala et al. (2014), Sadeghi et al. (2015) and Zhu et al. (2015), which use web information for the construction of the KS. Methods that use an external KS for a downstream task use a structured KS, such as the work of Narasimhan et al. (2018), Narasimhan & Schwing (2018), Wang et al. (2015) Wang et al. (2018) and Zhu et al. (2017). Zhu et al. (2017) introduced an iterative method for VQA tasks. Marino et al. (2019) introduced OK-VQA, a novel VQA dataset that requires the use of an external KS. Fan et al. (2020) applied a KS to multi-modal dialogue. In our work, we focus on a more natural KS, such as images and captions, which better reflect the data generated in newspapers and social media.

**Multi-modal Classification**    In this work, we investigate the potential advantages of using an external KS for the popular and challenging VQA domain, a multi-modal classification task. Current methods for VQA use pre-training on different datasets in order to gain better performance. In our experiments, we show performance for common methods such as VisualBERT (Li et al., 2019b), which concatenates the text and image modalities as an input to a pre-trained BERT (Devlin et al., 2018) model. ViLBERT (Lu et al., 2019) fuses text and image modalities using co-attentional transformer layers. Other methods such as Pythia (Jiang et al., 2018), VLBERT (Su et al., 2019) and MMBT (Kiela et al., 2019) can benefit from our method, as well as more recent work such as Oscar (Li et al., 2020b) and UNITER (Chen et al., 2020), which use the alignment task for pre-training their models. ~~The currently SOTA~~ In this paper, we choose to show our performance on the two common VisualBERT and ViLBERT models, and the winner of the VQA 2.0 2020 challange, Movie+MCAN  (Jiang et al., 2020) , which uses grid features instead of detection features, a modulated convolutional bottleneck for the image backbone, and MCAN (Yu et al., 2019) for fusion. A

Figure 1: (a) Cross-modal alignment architecture. We use a pre-trained ResNet-152 and BERT as feature extractors with an in-batch hinge loss. (b) Sample query image and retrieved captions from the COCO dataset. Ground truth captions are colored in blue (best viewed in color).

similar method was introduced by Nguyen et al. (2020). ~~Our method is also applicable to methods such as Pythia~~ (Jiang et al., 2018) ~~and MMBT~~ (Kiela et al., 2019).

## 3 METHOD

Our methodology is composed of two disjoint parts: (i) for a given external knowledge source $\mathcal{K}$, consisting of $m$ modalities, we train a model (*i.e.*, the *Retriever)* to align between the different modalities. (ii) Given a knowledge source $\mathcal{K}$ and an alignment model, we train a downstream model (*i.e.*, the *Reader)* by augmenting its inputs with extra data from $\mathcal{K}$.

### 3.1 CROSS-MODAL ALIGNMENT

Let $\mathcal{K}$ be a knowledge source consisting of $m$ modalities, where each sample $s_i = (s_i^0, \ldots, s_i^m) \in \mathcal{K}$ is a tuple of $m$ elements, corresponding to different modalities. Our alignment model encompasses $m$ encoders $E_m$, each composed of a feature-extraction module $F_m$, projection layer $P_m$, shared Transformer encoding layer $T$ with attention pooling, and a normalization layer $\mathcal{N}$:

$$E_m(x) = \mathcal{N}(T(P_m(F_m(x)))) \tag{1}$$

From this point, we will consider the two-modality case of images and captions as illustrated in Fig. 1. For text and image feature extractors, $F_1$ and $F_2$, we use a pre-trained BERT masked language model Devlin et al. (2018), and a pre-trained ResNet152 CNN backbone on ImageNet, respectively. The images are represented with convolutional grid features, chosen for robustness and speed, and these are flattened across the spatial dimension. The projection layers $P_m$ project each modality to a constant dimension $d$. The projected sequences are then forwarded to a shared Transformer-encoding layer, and aggregated by an attention pooling layer, resulting in a vector representation for each modality. Finally, we normalize each vector using an L2 normalizer, projecting all embeddings to the unit-sphere. Following Faghri et al. (2017), we only normalize the text embeddings because of image-caption imbalance (see Sec. 4.1).

We train our dense cross-modal retriever (DXR) using a contrastive loss, specifically using an in-batch hinge penalty with hard negatives (Faghri et al., 2017). Given a batch, consisting of $b$ samples, $s_1 \ldots s_b$, for each sample $s_i$, let $s_i^1$ and $s_i^2$ be the positive pairs and $s_i^1$ and $s_{j \neq i}^2$ the negative pairs. We compute the pair-wise similarity between the two modalities, using a dot product, denoted by $\pi(s_i^1, s_j^2) = \langle s_i^1, s_j^2 \rangle$. The hard-negative in-batch hinge loss is then defined as:

$$s_i^{2'} = \max_{j \neq i} \pi(s_i^1, s_j^2) \tag{2} \qquad\qquad s_i^{1'} = \max_{j \neq i} \pi(s_j^1, s_i^2) \tag{3}$$

$$\mathcal{L}_{hard} = \sum_i [\alpha + \pi(s_i^1, s_i^{2'}) - \pi(s_i^1, s_i^2)] + \sum_i [\alpha + \pi(s_i^{1'}, s_i^2) - \pi(s_i^1, s_i^2)] \tag{4}$$

where $s_i^{1'}$ and $s_i^{2'}$ are the hardest samples inside the batch, and $\alpha$ is the margin constant.

### 3.2 INDEXING AND RETRIEVING

Following Lewis et al. (2020), we use FAISS (Johnson et al., 2017) as our indexer platform for fast KNN queries. Given a knowledge source $\mathcal{K}$, we construct an index by computing the embeddings

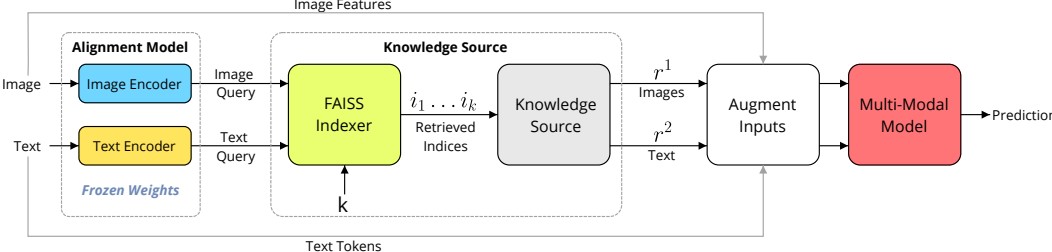

Figure 2: Illustration of our end-to-end framework. The trained cross-modal alignment is used to extract features as queries to a FAISS indexer. The $k$ retrieved indices are used to access data from the external knowledge source, and augment the input by appending each of the $k$ retrievals to the relative modality. For VQA, we only query the input image and retrieve $k$ captions.

of each sample in $\mathcal{K}$ using some alignment model (the *Retriever*), which can be trained on any arbitrary knowledge source. We introduce with two variants: we either construct separate indices $I_\mathcal{K}^m$ for each of the modalities; or we construct one joint index $I_\mathcal{K}$ that encompasses all modalities and where a KNN query will return a mixed modality result. Fig. 2 illustrates the two independent features of the alignment model and external knowledge source.

The retrieval process then consists of input query $q$, encoder $E_m$ and indexer $I_\mathcal{K}$ (or $I_\mathcal{K}^m$). $I_\mathcal{K}$ takes as an input an embedding query $e_q = E_m(q)$ and $k$, and returns the $k$-nearest indices $i_1 \ldots i_k$, corresponding to the $k$-nearest embeddings. We then index data from $\mathcal{K}$, resulting in $m$ retrieval sets $r^m = (r_1^m \ldots r_{n_m}^m)$, one for each modality, each consisting of varying number of samples $n_m$, where $\sum_{i=1}^m n_m = k$. When using $I_\mathcal{K}^m$, a single modality $m$ is returned, resulting in $r^m = (r_1^m \ldots r_k^m)$: For simplicity, we define the retriever by $R(q, E_m, I_\mathcal{K}, k) := \{r^1, \ldots, r^m\}$.

### 3.3 END-TO-END FUSION

Let $M$ be any multi-modal reader model, applied to a specific downstream task that takes as an input $x = (x^1, \ldots, x^m)$ consisting of $m$ modalities and outputs prediction $y$. The method augments the input $x$ by concatenating the retrieved samples to their corresponding input modalities, resulting in the augmented input $x'$:

$$x' = (x^1 \circ r_1^1 \circ \cdots \circ r_{n_1}^1, \ldots, x^m \circ r_1^m \circ \cdots \circ r_{n_m}^m) \tag{5}$$

The resulting end-to-end training of model $M$ is then defined by some loss function $\mathcal{L}$, minimizing $\mathcal{L}(M(x'), y)$, with the same hyperparameters as in the non-retrieval augmented case. Fig. 2 illustrates the complete model.

### 3.4 TIME-COMPLEXITY

As introduces in Sec. 2, we consider two types of retrievers, (i) methods such as ours, that use Maximum Inner Product Search (MIPS), where each modality is computed independently, and (ii) methods that have entangled computation of similarity between the different modalities, *e.g.*they cannot compute an independent embedding. Assuming a knowledge-source of size $N$, and a forward-pass with $O(1)$ time-complexity. In type (i), the embeddings of the entire knowledge source need to be computed only once, as well as the embeddings of the query sample. In our experiments, we use FAISS with "Hierarchical Navigable Small World" search, and as shown by Johnson et al. (2017), this searching method takes $O(AD\lceil \log N \rceil v)$, where $A$ and $v$ are constants, and $D$ is the degree of the graph. Therefore, the total time complexity of retrieving is $O(AD\lceil \log N \rceil v)$. On the other hand, methods of type (ii) must compute pairwise similarities between a query sample, and all samples in the dataset, resulting in a $O(N)$ run time for searching the most similar sample.

As a results, type (ii) methods are not applicable to our end-to-end fusion pipeline, where for each sample we query, a $O(N)$ is applied, which results in an inefficient and non-scalable method. On the other hand, our method does not impose significant overhead.

## 4 EXPERIMENTS

In this section, we describe the two experimental settings of the alignment model and the end-to-end downstream task training and evaluation. All models and experiments are implemented and performed with the MMF library (Singh et al., 2020a).

### 4.1 DATASETS

We use three common datasets for training and evaluating retrieval and VQA tasks. Flickr-30K (Young et al., 2014) is composed of 30,000 images, with 5 captions each. Following Karpathy & Fei-Fei (2015), we use 1000 images for validation and 1000 images for testing. COCO (Chen et al., 2015) is a well-known dataset that contains 120,000 images, with 5 captions each. We use the splits from Karpathy & Fei-Fei (2015) as well, resulting in 80K images for training, 5K images for validation and 5K images for testing. Following Faghri et al. (2017), we add an additional 30K images for training, and uses the same 1K and 5K splits. Conceptual Captions (Sharma et al., 2018) is a dataset that contains image-caption pairs, composed of 3M samples for training and 100K for validation, which we use to test our retrieval model.

The proposed datasets differ in two major axes (i) Size - The largest knowledge-source we use is CC, which contains 3M image-caption pairs, while COCO and Flickr30K are smaller in one and two orders of magnitude, respectively. (ii) Domain gap - As shown in (Singh et al., 2020b), CC datasets differs in both visual and textual domain from the VQA task, while COCO has the best domain match in both. Flickr30K datasets, on the other hand, is very similar to COCO, but suffers from short number of samples in an order of magnitude compared to COCO.

### 4.2 CROSS-MODAL RETRIEVAL

In the cross-modal retrieval task, we deal with two modalities: images and captions. Bi-directional retrieval is evaluated, denoted as Text $\rightarrow$ Image and Image $\rightarrow$ Text, where the left-hand-side indicates the query and the other indicates the retrieved domain. For fair comparison, we only report results for methods that use grid-features and vector representations, as noted in Sec 3.2 and 3.4. For a full comparison with other previous methods, please see Appendix A. Models are trained for 100K iterations with a warm-up of 2k iterations, batch size of 256, and Adam optimizer with learning-rate of 0.0001 where the (pre-trained unimodal) feature encoder's learning-rate is multiplied by 0.1. The hinge-loss margin hyperparameter $m$ is set to 0.2.

### 4.3 DOWNSTREAM TASKS

After training the alignment models for each dataset—Flickr30K, COCO and CC—we build indices for each, as defined in Sec 3.2. Note that for COCO, we only use the training set for indexing, while for Flickr30K and CC, we use the entire set of train/val/test. This is done for fair comparison with the VQA task, which uses the COCO training-set images for training. Our experiments focus on VQA as the downstream task, however we note that extension to other multi-modal tasks is straightforward. The inputs of the VQA task are image and text tuples, and it is presented as a classification problem over a set of answers. In VQA, we observe that information regarding the content of the image, such as the amount, color and location of objects is very correlated with the question and answer. Therefore, captions serve as good auxiliary information, while similar/retrieved images (e.g., to which the question does not directly refer) are less informative. For that reason, we use the *separate indices* variant, retrieving text captions from images to yield a cross-modal image to text translation. We experiment with all three datasets, evaluating different training and inference variants.

## 5 RESULTS

### 5.1 CROSS-MODAL RETRIEVAL

Tab. 1 and 2 show retrieval results on Flickr30K and COCO, comparing similar methods that use grid-features and vector representations for the embedding space. Reported numbers correspond to Recall-at-1/5/10 on the test-sets. As can be seen, our method significantly outperforms previous

| Method | Text → Image | | | Image → Text | | |
|---|---|---|---|---|---|---|
| | R@1 | R@5 | R@10 | R@1 | R@5 | R@10 |
| RRF | 35.4 | 68.3 | 79.9 | 47.6 | 77.4 | 87.1 |
| CMPM | 37.3 | 65.7 | 75.5 | 49.6 | 76.8 | 86.1 |
| DPC | 39.1 | 69.2 | 69.2 | 55.6 | 81.9 | 89.5 |
| VSE++ | 39.6 | 69.6 | 79.5 | 52.9 | 79.1 | 87.2 |
| **DXR** | **50.6** | **78.8** | **86.7** | **65.1** | **87.3** | **92.6** |

Table 1: Retrieval results for Flickr30K, comparing only methods that use raw images as input, and vector representations for the embedding space. Additional methods can be found in Appendix A.

| | COCO 1K | | | | | | COCO 5K | | | | | |
|---|---|---|---|---|---|---|---|---|---|---|---|---|
| | Text → Image | | | Image → Text | | | Text → Image | | | Image → Text | | |
| Method | R@1 | R@5 | R@10 | R@1 | R@5 | R@10 | R@1 | R@5 | R@10 | R@1 | R@5 | R@10 |
| DPC | 47.1 | 79.9 | 90.0 | 65.6 | 89.8 | 95.5 | 25.3 | 53.4 | 66.4 | 41.2 | 70.5 | 81.1 |
| VSE++ | 52.0 | 83.1 | 92.0 | 64.6 | 89.1 | 95.7 | 30.3 | 59.1 | 72.4 | 41.3 | 69.2 | 81.2 |
| CMPM | 44.6 | 78.8 | 89.0 | 56.1 | 86.3 | 92.9 | 22.9 | 50.2 | 63.8 | 31.1 | 60.7 | 73.9 |
| **DXR** | **56.8** | **88.2** | **94.9** | **67.0** | **93.0** | **97.6** | **33.9** | **64.9** | **77.4** | **44.9** | **75.2** | **84.7** |

Table 2: Retrieval results for COCO, comparing only methods that use raw images as input, and vector representations for the embedding space. Additional methods can be found in Appendix A.

| Knowledge Source | Training Type | Visual BERT | ViLBERT | Movie+MCAN |
|---|---|---|---|---|
| Flickr30K | XTRA 10-C | 66.77 | 67.32 | 69.70 |
| CC | PT | 64.34 | 68.14 | - |
| | XTRA-10C | 67.49 | 67.37 | 69.02 |
| | PT + XTRA-10C | 67.53 | 69.17 | - |
| COCO | PT | 64.54 | 67.58 | - |
| | XTRA-10C | **68.98** | 69.07 | **71.52** |
| | PT + XTRA-10C | 67.71 | **69.90** | - |
| | Vanilla | 63.54 | 67.56 | 71.16 |
| | 5-GT | 69.61 | 71.50 | 71.80 |

Table 3: VQA Results for three different reader models on COCO `val-set`. **Vanilla** - models use pre-trained BERT model. **PT** - Pre-Training with the knowledge source. **XTRA-10C** - training via our method using the knowledge source indicated and alignment model trained on that knowledge source, using 10 retrieved captions. **5-GT** - training with the 5 ground truth captions.

work in both datasets. We refer to Appendix A for a comparison with additional methods. While CC is not commonly used in the retrieval literature, we use it for our downstream task. Using DXR, we obtain the following results for CC: R@1: 25.1 R@5: 50.1 and R@10: 61.9 for Text → Image, and R@1: 25.4 R@5: 50.9 and R@10: 61.8 for Image → Text. The alignment model trained on CC is used for training in the downstream VQA task. We notice that performance degrades as the dataset size increases, which could affect the downstream task since we query from the entire dataset.

## 5.2 VISUAL QUESTION ANSWERING

Our main results show performance on the VQA `val-set`, experimenting with three common VQA methods: VisualBERT (Li et al., 2019b), ViLBERT (Lu et al., 2019), and the currently ~~state-of-the-art~~ winner of the VQA 2.0 challenge, Movie+MCAN (Nguyen et al., 2020), each along with three different knowledge sources (COCO, CC and Flickr30K). Following Jiang et al. (2020), we use the `val-set` to assist in our exhaustive ablation studies, however we report our final ~~SOTA~~ result on the VQA `test-std` split. Tab. 3 summarizes four different training settings: (i) **vanilla**

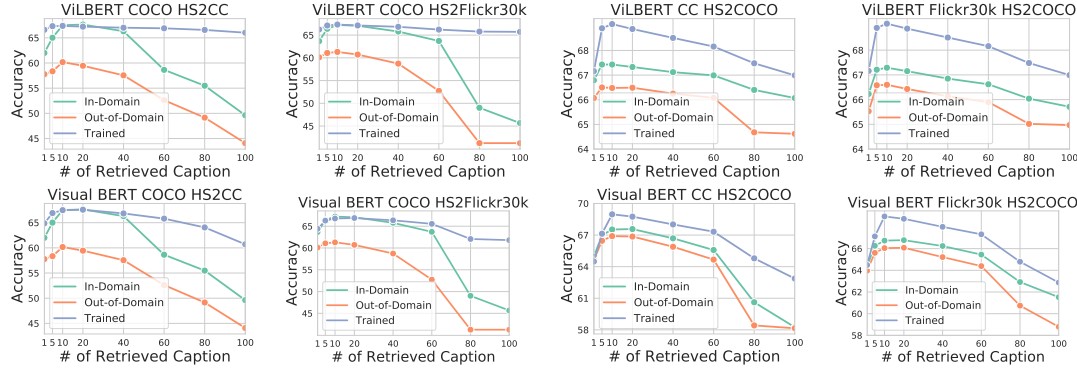

(a)                                                (b)

Figure 3: Two Hot-Swap configurations of the knowledge source during inference. **(a)** both the alignment model and the knowledge source are replaced with new ones built using a new dataset. **(b)** only the knowledge source is replaced, and the indexer is built using the old alignment model.

Figure 4: Hot-Swap results. Each row corresponds to a different reader model. Each graph shows **(a)** Training with different amount of retrieved captions. **(b)** Using the trained model with 10-cap, we inference with different amount of captions. **(c)** Hot swapping between knowledge sources.

- models using pre-trained BERT; (ii) **PT** - Task agnostic pre-Training with the knowledge source dataset (using masked language modeling); (iii) **5-GT** - training with the 5 ground truth captions from COCO; (iv) **XTRA-10C** - training via our method, using the knowledge source indicated and alignment model trained on that source, using 10 retrieved captions.

We see that using the five ground truth (GT) COCO captions as additional data (bottom row of Tab. 3), sets a soft upper bound for our approach. On one hand, the GT captions contain relevant information about the content of the image; on the other hand, other captions from the knowledge source may additionally serve as rich, useful descriptions. We also see that our method significantly increases performance across all baselines, even with respect to pre-training. This suggests that our method serves as a good alternative for pre-training. Our best model, Movie+MCAN+XTRA-10C, outperform Movie+MCAN on *VQA2.0* `test-std`, ~~Our best model sets a new state-of-the-art on~~ *VQA2.0* ~~`test-std`, using Movie+MCAN+XTRA-10C,~~ obtaining a score of **73.12** (with single model performance).

## 5.3 HOT SWAP

Our method is devised such that querying and retrieving from the knowledge source is independent of the downstream model, enabling the swap of the alignment model and/or knowledge source during inference. This affords interesting explorations. We describe two forms of "hot swapping": (i) the entire knowledge source with its trained alignment model are replaced with a new one and corresponding alignment model – we refer to this as "out-of-domain"; (ii) the knowledge source used for retrieving is swapped, but the alignment model remains the same as was originally trained with the downstream model. In this case, we build a new retriever for the new knowledge source, using the original alignment model – we call this "in-domain". "in/out-of-domain" refers to the alignment domain with which the downstream model was trained. Fig. 3 illustrates the two cases.

In Fig. 4 we show different inference results for hot swapping. All models in this experiment are trained using 10 retrieved cations. The title of each graph represents the trained model, followed by the trained knowledge source and the knowledge source to which we swap. In addition, we show inference results for training with the swapped knowledge source, *e.g.* training with CC knowledge source and alignment model from scratch, using 10 retrievals. As can be seen, "in-domain" hot

| Query Image | | Retrieved Captions | |
| --- | --- | --- | --- |
| | No Hotswap | Flickr30K Hotswap | CC Hotswap |
| COCO `val-set` | COCO `train-set` | `train+val+test sets` | `train+val sets` |
| | A dog that is lying down on a sidewalk | A dog asleep on the streets | A dog lies down on a cobblestone street |
| | A dog with a muzzle on is lying on the sidewalk | A tan male bulldog sleeping on a sidewalk | The dog is lying on the cobblestone street |
| | A happy stray puppy lies in the street | Cute dog sleeping on the sidewalk | A dog laying on the side of the street |
| | A dog is laying and resting on a walkway | A dog lying on the sidewalk | A dog with a collar on lying on the street |

Figure 5: Sample top-4 result for "in-domain" Hot-Swap. The model was trained using COCO as the knowledge source, and 10 retrieved captions. Left - Query image from VQA `val-set`. Columns refer to the different hot-swaps, showing retrieved captions.

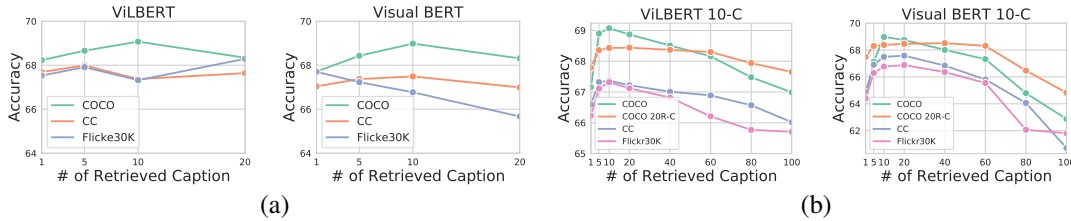

(a)                                                          (b)

Figure 6: Ablation study of our method. **(a)** - Training with different amount of retrieved captions. **(b)** - Using the trained model with 10-cap, we inference with different amount of captions.

swapping performance is significantly higher than "out-of-domain". We hypothesize that the reader model has learned an implicit structure of the alignment space. Surprisingly, when training with COCO as the knowledge source, "in-domain" hot swapping performs similarly, for the same amount of trained retrievals (10), as training with an alternative knowledge source and alignment model. On the other hand, we observe that this suffers from a decrease in generalization due to different amounts of retrieval during inference-time. In the other direction, hot swapping to COCO from CC or Flickr30K does not result in the same performance as training with COCO as the knowledge source and alignment model, yet, performance and generalization do not degrade. Qualitative results of "in-domain" hot swapping are presented in Fig 5. As can be observed, novel information such as the fact that the image shows a "cobblestone street" is retrieved from CC without having to train the alignment model on that source.

## 5.4   ABLATION STUDY

In this study, we explore the use of different amounts of retrieval during training and inference, as well as doing inference without retrieving - which we name *unplugged*. We further explore the relationship between pre-training and XTRA.

**Number of Retrievals**    We experiment with different amounts of retrieved captions during training and inference. In Fig 6 (a), we show the performance of our method when training with different amounts of retrieval, and different knowledge sources. As can be observed, training with 10 captions and COCO as the knowledge source results in the best performance. In Fig 6 (b), we show the inference performance for models trained using 10 retrievals. In addition, we show the inference performance of the same model, trained with random amounts of retrieval, between 1 and 20, on the COCO dataset (COCO 20R-C). With this, the best performance is given when we inference with the same amount of trained retrievals, and this then degrades as the number of retrievals differ from how the model was trained. We also see that training with varying number of retrievals achieves better generalization to different amounts of retrievals during inference, as can be seen in Fig 6 (b), COCO 20R-C, where performance is maintained up to 60 retrievals during inference.

**Unplugged Performance**    One interesting observation we make is the ability to "unplug" the knowledge source by not retrieving during inference-time. Tab. 4 shows a noticeable decrease in performance, indicating the dependency of the reader on the retrieved data during training. When training with COCO as the knowledge source, introducing captions that are very related to the input images is biasing the model to depend on the retrieved captions. For CC and Flickr30K, the domain

| Model | COCO | CC | Flickr30K |
|---|---|---|---|
| Visual BERT | 58.77 (68.98) ↓ 10.21 | 63.15 (67.49) ↓ 4.34 | 61.86 (66.77) ↓ 4.91 |
| ViLBERT | 45.60 (69.07) ↓ 23.47 | 63.50 (67.37) ↓ 3.87 | 59.34 (67.32) ↓ 7.98 |

Table 4: VQA performance using models trained with 10 retrieved caption, and evaluating without any retrievals ("unplugged"). The highest decrease in performance occurs for the in-domain (COCO) knowledge source where retrieved examples are most informative.

gap between the downstream task and the knowledge source lessens this gap in unplugged performance. Surprisingly, while ViLBERT performance is generally better than Visual BERT, using our method, the opposite is true when *unplugging* the knowledge source.

**External Knowledge Source & Pre-training** The use of a retrieval mechanism over external knowledge sources raises inriguing questions, e.g.: 1) Is augmentation better than pre-training?; and 2) Can pre-training help the external knowledge source? To address these questions, we experimented with two different pre-training datasets commonly used for VQA - COCO and CC. Tab. 3 suggests that for the COCO dataset, using our method is significantly better than pre-training alone, while using pre-training followed by XTRA causes the performance to vary with respect to the reader architecture (e.g., pre-training helps XTRA with ViLBERT, but not with VisualBERT). Tab. 3 also shows that fine-tuning our method after pre-training on the same knowledge source yields better performance over pre-training across all knowledge sources and architectures.

## 6 CONCLUSION

In this work, we presented a novel approach that proposes the use of external knowledge sources in multi-modal prediction models with transformer architectures. We trained a state-of-the-art alignment model, DXR, for performing retrieval over external knowledge sources. We showed that our method XTRA yields gains in performance when using an in-domain knowledge source on VQA. We conducted a variety of experiments to show the sensitivity and effects of the used knowledge source with various choices in hyperparameters which shed further light on the different aspects of the model. Future research and applications of our method include interpretability of retrieved data and predictions for verification processes, the demonstration of increased information security by hot-swapping, and unplugged versions of models and new architectures that take advantage of out-of-domain knowledge source. We hope that our approach inspires further work in the direction of hybrid parametric non-parametric models for multi-modal problems.

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

| Method | Text → Image | | | Image → Text | | |
|---|---|---|---|---|---|---|
| | R@1 | R@5 | R@10 | R@1 | R@5 | R@10 |
| RRF | 35.4 | 68.3 | 79.9 | 47.6 | 77.4 | 87.1 |
| CMPM | 37.3 | 65.7 | 75.5 | 49.6 | 76.8 | 86.1 |
| DPC | 39.1 | 69.2 | 69.2 | 55.6 | 81.9 | 89.5 |
| VSE++ | 39.6 | 69.6 | 79.5 | 52.9 | 79.1 | 87.2 |
| **DXR** | **50.6** | **78.8** | **86.7** | **65.1** | **87.3** | **92.6** |
| TERN | 41.1 | 71.9 | 81.2 | 53.2 | 79.4 | 86.0 |
| SCO | 41.1 | 70.5 | 80.1 | 55.5 | 82.0 | 89.3 |
| SAEM | 52.4 | 81.1 | 88.1 | 69.1 | 91.0 | 95.1 |
| SCAN | 48.6 | 77.7 | 85.2 | 67.4 | 90.3 | 95.8 |
| CAMP | 51.5 | 77.1 | 85.3 | 68.1 | 89.7 | 95.2 |
| VSRN | 54.7 | 81.8 | 88.2 | 71.3 | 90.6 | 96.0 |
| TERAN | 56.5 | 81.2 | 88.2 | 70.8 | 90.9 | 95.5 |
| MMCA | 54.8 | 81.4 | 87.8 | 74.2 | 92.8 | 96.4 |
| Unicoder-VL | 71.5 | 90.9 | 94.9 | 86.2 | 96.3 | 99.0 |
| UNITER | 73.6 | 93.0 | 95.9 | 88.2 | 98.4 | 99.0 |

Table 5: Retrieval results for Flickr30K. **Top** - methods that use raw images as input, and vector representations for the embedding space. **Bottom** Methods that use detection features or sequence similarity measures.

| | COCO 1K | | | | | | COCO 5K | | | | | |
|---|---|---|---|---|---|---|---|---|---|---|---|---|
| | Text → Image | | | Image → Text | | | Text → Image | | | Image → Text | | |
| Method | R@1 | R@5 | R@10 | R@1 | R@5 | R@10 | R@1 | R@5 | R@10 | R@1 | R@5 | R@10 |
| DPC | 47.1 | 79.9 | 90.0 | 65.6 | 89.8 | 95.5 | 25.3 | 53.4 | 66.4 | 41.2 | 70.5 | 81.1 |
| VSE++ | 52.0 | 83.1 | 92.0 | 64.6 | 89.1 | 95.7 | 30.3 | 59.1 | 72.4 | 41.3 | 69.2 | 81.2 |
| CMPM | 44.6 | 78.8 | 89.0 | 56.1 | 86.3 | 92.9 | 22.9 | 50.2 | 63.8 | 31.1 | 60.7 | 73.9 |
| **DXR** | **56.8** | **88.2** | **94.9** | **67.0** | **93.0** | **97.6** | **33.9** | **64.9** | **77.4** | **44.9** | **75.2** | **84.7** |
| TERN | 51.9 | 85.6 | 93.6 | 63.7 | 90.5 | 96.2 | 28.7 | 59.7 | 72.7 | 38.4 | 69.5 | 81.3 |
| SCO | 56.7 | 87.5 | 94.8 | 69.9 | 92.9 | 97.5 | 33.1 | 62.9 | 75.5 | 42.8 | 72.3 | 83.0 |
| SAEM | 57.8 | 88.6 | 94.9 | 71.2 | 94.1 | 97.7 | - | - | - | - | - | - |
| SCAN | 58.8 | 88.4 | 94.8 | 72.7 | 94.8 | 98.4 | 38.6 | 69.3 | 80.4 | 50.4 | 82.2 | 90.0 |
| CAMP | 58.5 | 87.9 | 95.0 | 72.3 | 94.8 | 98.3 | 39.0 | 68.9 | 80.2 | 50.1 | 82.1 | 89.7 |
| VSRN | 62.8 | 89.7 | 95.1 | 76.2 | 94.8 | 98.2 | 40.5 | 70.6 | 81.1 | 53.0 | 81.1 | 89.4 |
| TERAN | 65.0 | 91.2 | 96.4 | 77.7 | 95.9 | 98.6 | 42.6 | 72.5 | 82.9 | 55.6 | 83.9 | 91.6 |
| MMCA | 61.6 | 89.8 | 95.2 | 74.8 | 95.6 | 97.7 | 38.7 | 69.7 | 80.8 | 54.0 | 82.5 | 90.7 |
| Unicoder-VL | 69.7 | 93.5 | 97.2 | 84.3 | 97.3 | 99.3 | 46.7 | 76.0 | 85.3 | 62.3 | 87.1 | 92.8 |
| UNITER | - | - | - | - | - | - | 51.7 | 78.4 | 86.9 | 66.6 | 89.4 | 94.2 |
| Oscar | 78.2 | 95.8 | 98.3 | 89.8 | 98.8 | 99.7 | 57.5 | 82.8 | 89.8 | 73.5 | 92.2 | 96.0 |

Table 6: Retrieval results for COCO. **Top** - methods that use raw images as input, and vector representations for the embedding space. **Bottom** Methods that use detection features or sequence similarity measures.

# A    RETRIEVAL

Tab. 6 shows a complete comparison of the different alignment methods in the cross-modal alignment literature. The top part corresponds to methods which use vector representations, grid-features, and do not share information between the modality branches. The bottom part shows the rest of the methods.

