# OpenReview forum: "Cross-Modal Retrieval Augmentation for Multi-Modal Classification"
_ICLR.cc/2021/Conference — Reject_

### Official Review · AnonReviewer3 · 2020-10-27
**The advantages and significance of the proposed method are unclear and confusing.**

**Rating:** 5
**Confidence:** 5

**Review:**

In this paper, the authors present a method to use unstructured external knowledge sources to improve visual question answering and image-caption retrieval. The proposed method can achieve somewhat improvement for visual question answering, but drop the performance for image-caption retrieval with a more complex model. Some concerns are as follows:

1. The authors claim that the proposed method achieved state-of-the-art performance on both COCO and Flickr30k  image-caption retrieval. However, their retrieval scores are lower about 10 than the state-of-the-art counterparts, such as TERAN. The statement is not correct.
2. Although the authors stated the proposed method uses raw images as input, the adopted backbones (i.e., image/text encoders) should be frozen to extract the features for the following components in their pipeline, which is similar to the other feature-based methods (e.g., TERAN) that also can be seen as freezing their backbones (e.g., Faster R-CNN) during their training and inference stages. Thus, the inputs between the proposed method and other methods have no essential difference. What is the significance to design such a much more complex model for image-caption retrieval? What are the advantages of the proposed method comparing prior superior methods? I am confused that if it is worthy to adopt such a complex model with worse performance.
3. It is interesting to see that the proposed method could improve the performance of VQA. However, Table 3 does not give us a throughout comparison. There are many results missed in the table, such as different training types for Flickr30K, some results for Movie+MCAN, etc. From the results, we also could draw that the improvement of the proposed method is very limited for a good VQA method, i.e., Movie+MCAN with Vanilla. The experiments could not significantly demonstrate the significance and advantages of the proposed method.

---

> ### Author Response · Authors · 2020-11-20
> **Response to Reviewer3**
>
> Thank you for the detailed review.
>
> We kindly point the reviewer to our “general response” for detailed explanation of major concerns.
>
> ### Concern 1+2 - DXR performance
>
> 1) **Q:** The reviewer is concerned with the performance of our retriever, with respect to other baselines.
>
>    **A:** As we mention in the related work section, there are two types of retrievers. Type (i), such as our DXR, enjoy an $O(log{N})$ time-complexity for a single query, assuming a knowledge source of size $N$. Resulting in applicable methods for the downstream tasks. Type (ii) methods, which perform better on the cross-modal retrieval task, suffer from high - $O(N)$ time-complexity for a single query, which is not applicable for the proposed integration with a downstream task. We will make sure to clearly state the differences.
>
> 1) **Q:** The reviewer is concerned with the different inputs in the training pipeline. Where for some methods, detection features are used for the VQA model, and this is similar to retriever models such as TERAN. “ What is the significance to design such a much more complex model for image-caption retrieval? What are the advantages of the proposed method comparing prior superior methods?”
>
>    **A:** We choose to use the raw image as the input to our retriever following recent advancements in VQA methods such as Movie+MCAN, where the input to their model is the raw image as well. Therefore, our method does not enforce additional complexity in that sense.
> Regarding methods such as TERAN - these are methods that suffer from high computational-complexity for retrieval, and are not applicable for the downstream task as we stated above.
> Regarding the complexity of our retriever (DXR) with respect to TERAN - we could not find any significant difference in the complexity of the models. We kindly ask the reviewer to elaborate on this subject.
>
> ### Concern  - VQA performance
>
> 1) **Q:** “Table 3 does not give us a throughout comparison. There are many results missed in the table”
>
>    **A:** For fair comparison, we follow common training procedures, meaning, Visual BERT and ViLBERT use COCO and CC for pre-training, while Movie+MCAN does not perform any pre-training. Our method improves the given baselines, while not changing their training scheme. We believe this point is in our favor, where we try to remain agnostic to the model we try to improve. Indeed, as models perform better, the gap in performance decreases, but as can be seen, our method introduces additional capabilities which we hope to be applied in future work such as interpretability and verification of models.

---

### Official Review · AnonReviewer2 · 2020-10-29

**Rating:** 4
**Confidence:** 5

**Review:**

This paper explores a new direction, to utilize the searched results (image caption pair) to improve downstream multimodal learning tasks. They first pre-trained a cross-modal model using the contrastive learning on the image caption dataset. Then they use the pre-trained model to search the relevant terms for image or text input, and augment the searched results as the input for the downstream multi-modal tasks.

Concerns:

1. The authors claim that the trained alignment model, DXR, "achieves state-of-the-art image-caption retrieval performance" on COCO and MIRFlickr. However, plenty of cross-modal retrieval methods achieve better performance than DXR. For example, Oscar [a] and Unicoder-VL [b] could achieve significantly better than DXR on both COCO and MIRFlickr. In my mind, this could not be regarded as a novel contribution. Also, how different cross-modal pre-trained models perform as the DXR is also an interesting direction for exploration.

2. The authors claim that XTRA "achieved state-of-the-art performance", which is also not convincing. Oscar [a] and UNITER [c] achieve significantly better than XTRA on COCO VQA val set. Please carefully survey the literature and claim the contribution.

3. The visual-linguistic pre-training methods [a,b,c] also aim to distill the external knowledge from the large-scale image caption dataset to the downstream tasks, but the solution is to provide a pre-trained model as the initialization which is pre-trained on the large scale image-caption dataset. These methods should also be surveyed in the literature, and may be compared in the experiment part.

[a] Oscar: Object-Semantics Aligned Pre-training for Vision-Language Tasks, ECCV 2020

[b] Unicoder-VL: A Universal Encoder for Vision and Language by Cross-modal Pre-training, AAAI 2020

[c] UNITER: UNiversal Image-TExt Representation Learning, ECCV 2020

---

> ### Author Response · Authors · 2020-11-20
> **Response to Reviewer2**
>
> Thank you for the detailed review.
>
> We kindly point the reviewer to our “general response” for detailed explanation of major concerns.
>
> ### Concern 1 - DXR performance
>
> 1) **Q:** The reviewer points to additional methods that achieve better performance than DXR, and is concerned about the method performance.
>
>    **A:** We state in the related work section, that we distinguish between two different types of retrievers. The major difference is in the computational-complexity, where methods such as pointed by the reviewer require the computation of all pairwise similarities for a novel sample, which render these methods inapplicable to the downstream task. For fair comparison, we compare with similar methods to ours. We will make sure to add missing methods to the relevant tables.
>
>
> 2) **Q:** “how different cross-modal pre-trained models perform as the DXR is also an interesting direction for exploration.”
>
>    **A:** We thank the reviewer for mentioning DXR as an interesting direction. As DXR is shown to be significantly better than similar methods, a degraded retriever for the downstream task can be seen as using a knowledge source with domain shift w.r.t. the VQA dataset. Such an example can be seen when using CC and Flickr as knowledge sources, where performance is lower.
>
> ### Concern 2+3 - XTRA performance
>
> 1) **Q:** The reviewer is concerned with the performance of the proposed XTRA method, and points to recent methods that achieve better performance.
>
>    **A:** We apologize for missing Oscar and UNITER. Truly, they perform better than what we present in the paper. We will retract the claim for SOTA.
> Our method shows an improvement over the method proposed by the winners of the VQA 2.0 2020 challenge. We will make sure to properly address our contribution, where we show improvement given three different baselines.

---

### Official Review · AnonReviewer1 · 2020-11-02
**Interesting paper, but the proposed method is not novel enough and problems in experiment comparison.**

**Rating:** 3
**Confidence:** 5

**Review:**

This paper proposed a cross-modal retrieval augmentation for the multi-modal classification task (VQA). The authors first introduce a transformer-based image caption retrieval architecture that achieves decent performance. Then, the authors proposed to use the retrieval model to retrieve relevant visual and textual information as augmentation. The proposed method experiment on 3 existing method (Visual Bert, ViLBERT, and Movie + MCAN) and show good improvements over the baseline model.

My major concern about this paper is the lack of novelty and experiment comparison. The proposed image caption retrieval architecture is not novel at all. Most existing method (ViLBERT, UNITER, VLBERT etc.) has a similar transformer objective, while the image fine-tuning are from Pixel Bert. In the experiment section (Table 2), the author even didn't compare these methods.

In terms of the VQA performance, pre-training on the conceptual caption actually hurt the performance of visual Bert and ViLBERT. Could the authors explain why a larger dataset can not help with the model? Is Cross-Modal Retrieval pre-training necessary for VQA?

In terms of the speed, the deep fusion model for image retrieval is super slow, since the model need to calculate the score for each pair. What is the size of pool when computing the retrieved captions? What is the time complexity?

---

> ### Author Response · Authors · 2020-11-20
> **Response to Reviewer1**
>
> Thank you for pointing out your concerns so clearly.
>
> We kindly point the reviewer to our “general response” for detailed explanation of major concerns.
>
> ### Cross-Modal retrieval & Time-Complexity:
>
> 1) **Q:** The reviewer is concerned about the paper’s novelty and comparisons to other works. Additional methods are pointed by the reviewer to achieve higher performance. The reviewer is also concerned about the time complexity of the method.
>
>    **A:** While methods proposed by the reviewer achieve better performance, they are not applicable to the downstream task, due to their high computational-complexity in finding the most similar sample in the knowledge source. These methods need to compute the entire pairwise similarity to an input sample. Our method on the other hand, computes each embedding independently, and performs a fask KNN search to find the most similar sample. Thus, for fair comparison, we explicitly distinguish between the different methods
>
> 2) **Time-Complexity analysis** - Assuming we use a knowledge-source of size $N$, and applying a forward-pass takes $O(1)$. The following are the time-complexity for each retrieval type during inference, for a single sample query:
>    * **Type (i)** - We use “Hierarchical Navigable Small World” search, which has time-complexity of $O(AD[\log N]v)$ (Johnson et al., 2017), where $A$ and $v$ are constants, and $D$ is the degree of the graph. Therefore, the total time complexity of retrieving is $O(AD[\log N]v)$.
>     * **Type (ii)** - Because these methods must compute pairwise similarities, the total time complexity of retrieving is $O(N)$
>
> ### VQA:
>
> We would like to ask the reviewer for more clarifications regarding the noted concerns. To the best of our understanding, these are the concerns pointed by the reviewer:
> 1) **Q:** The reviewer is concerned about the performance of pre-training with “conceptual captions”, which they points out as hurting the performance of visual Bert and ViLBERT
>
>    **A:** We assume the reviewer points to Tab.3, where pre-training on CC is lower than our method. Otherwise, we see that pre-training on “conceptual captions” is actually better than Vanilla, and for ViLBERT is also better than pre-training on COCO
>
> 2) **Q:** Does Cross-Modal Retrieval pre-training necessary for VQA?
>
>    **A:** Our method does not apply pre-training on cross modal retrieval. It trains a retriever on a knowledge source, and uses it to retrieve additional inputs, as we describe throughout the paper.
> We also note that using two pre-trained unimodals (such as ResNet and BERT) to extract embeddings does not work, because the embedding space was not trained to align the two modalities. We will clarify this in the text
>
> 3) Regarding the gap in performance between COCO Flickr and CC - It has been shown by [a],  that CC dataset differ in both visual and textual domain from the VQA task, while COCO has the best domain match in both. Flickr30K datasets, on the other hand, is very similar to COCO, but suffers from a short number of samples in an order of magnitude compared to COCO. We will clarify this in the text
>
> [a] Singh et. al. “Are we pretraining it right? Digging deeper into visio-linguistic pretraining”, 2020

---

### Author Response · Authors · 2020-11-20
**Clarifications of the proposed method**

We thank the reviewers for their constructive comments. In hindsight, we think that some aspects of our proposed method were not explained as well as they should have been, causing some confusion. In this general response we address comments that are shared across the reviewers:

### Cross-Modal retrieval : Comparison to previous work and time-complexity

* The Cross-modal retriever is trained on each knowledge source, and later used for retrieval in the downstream task, and is not learned during the training of the downstream task. Therefore, we do not pre-train on the cross-modal alignment task, as in Oscar[1] for example. The downstream tasks model are trained from scratch using the proposed method.
* **In the paper we explicitly distinguish between two types of methods** (“Related Work” - “Cross-Modal Retrieval”): (i) methods that use grid-features and/or vector representations of the embedding space, and (ii) methods that use detection features, sequence representations, or share information between the two modalities for computing the similarity metric.
* The reason we distinguish between the two types of retrieval methods is due to Time-Complexity and applicability to be integrated in the downstream task:
  * **Methods of type (i)** , such as ours have the following properties:
    1) Each modality is computed independently of the other modalities
    2) For each dataset we compute the embedding only once
    3) We use MISP to perform fask KNN, and retrieve data from the knowledge source. For that, we only compute the embeddings of a new sample, and the run-time is about $O(\log N)$, where $N$ is the knowledge source size
  * **Methods of type (ii)**, such as proposed by the reviewers (Oscar[1], UNITER[2]) suffer from:
    1) They suffer from an entangled computation of modals similarities, e.g. they cannot compute an independent embedding
    2) In order to retrieve samples, one needs to compute pairwise similarities between a query sample, and all samples in the dataset, which results in an $O(N)$ run-time, where $N$ is the knowledge source size.
    3) Though such models enjoy high performance of the evaluation benchmarks, they are not applicable to the proposed downstream task, because of their inefficiency
    4) Methods suggested by reviewers will be properly addressed and add to the paper
* We will add a section concerning these differences, and discussing the time complexity of each type

### XTRA performance

* We apologize for missing the comparison to Oscar[1] and UNITER[2]. Indeed, their performance on test-std is better than what we report in the paper. We note that these are both recent papers (ECCV happened 3 months ago), but still. We will retract the claim for SOTA, and properly address those works
* As our method is modular and can be applied within any given model, we chose to experiment with the method proposed by the winners of the VQA 2.0 2020 challenge, and show improvement over their performance when bounding training to COCO (no additional VQA data is used for training)
* Our contribution is the training scheme, which involves the retriever and an external knowledge source. We will make sure to properly address the contribution and performance with respect to previous work, where we show an improvement over the proposed baselines

[1] Oscar: Object-Semantics Aligned Pre-training for Vision-Language Tasks, ECCV 2020
[2] UNITER: UNiversal Image-TExt Representation Learning, ECCV 2020

---

### Decision · Program_Chairs · 2021-01-07
**Final Decision**

**Decision:**

Reject

**Comment:**

The paper discusses the problem of how to augment cross-modal retrieval for the task of multi-modal classification -- it uses image caption pairs to improve downstream multimodal learning, and shows improvement in the task of visual question answering. However, the paper has the following weaknesses: (a) lack of novelty, (b) lack of thorough empirical evaluation, (c) the complex model did not give significant gains.